# In Vivo Tracking for Oncolytic Adenovirus Interactions with Liver Cells

**DOI:** 10.3390/biomedicines10071697

**Published:** 2022-07-13

**Authors:** Victor A. Naumenko, Daniil A. Vishnevskiy, Aleksei A. Stepanenko, Anastasiia O. Sosnovtseva, Anastasiia A. Chernysheva, Tatiana O. Abakumova, Marat P. Valikhov, Anastasiia V. Lipatova, Maxim A. Abakumov, Vladimir P. Chekhonin

**Affiliations:** 1V. Serbsky National Medical Research Center for Psychiatry and Narcology, 119034 Moscow, Russia; logannaridenna@gmail.com (D.A.V.); a.a.stepanenko@gmail.com (A.A.S.); sollomia@yandex.ru (A.O.S.); aachernysheva512@gmail.com (A.A.C.); marat.valikhov@gmail.com (M.P.V.); chekhoninnew@yandex.ru (V.P.C.); 2Department of Medical Nanobiotechnology, N.I Pirogov Russian National Research Medical University, 117997 Moscow, Russia; abakumov1988@gmail.com; 3Skolkovo Institute of Science and Technology, Bolshoy Boulevard 30, 121205 Moscow, Russia; sandalovato@gmail.com; 4Center for Precision Genome Editing and Genetic Technologies for Biomedicine, Engelhardt Institute of Molecular Biology, Russian Academy of Sciences, 119991 Moscow, Russia; lipatovaanv@gmail.com; 5Laboratory of Biomedical Nanomaterials, National University of Science and Technology (MISIS), 119049 Moscow, Russia

**Keywords:** adenovirus, liver, hepatotoxicity, zeiosis, neutrophils, CD8+ T cells, intravital microscopy

## Abstract

Hepatotoxicity remains an as yet unsolved problem for adenovirus (Ad) cancer therapy. The toxic effects originate both from rapid Kupffer cell (KCs) death (early phase) and hepatocyte transduction (late phase). Several host factors and capsid components are known to contribute to hepatotoxicity, however, the complex interplay between Ad and liver cells is not fully understood. Here, by using intravital microscopy, we aimed to follow the infection and immune response in mouse liver from the first minutes up to 72 h post intravenous injection of three Ads carrying delta-24 modification (Ad5-RGD, Ad5/3, and Ad5/35). At 15–30 min following the infusion of Ad5-RGD and Ad5/3 (but not Ad5/35), the virus-bound macrophages demonstrated signs of zeiosis: the formation of long-extended protrusions and dynamic membrane blebbing with the virus release into the blood in the membrane-associated vesicles. Although real-time imaging revealed interactions between the neutrophils and virus-bound KCs within minutes after treatment, and long-term contacts of CD8+ T cells with transduced hepatocytes at 24–72 h, depletion of neutrophils and CD8+ T cells affected neither rate nor dynamics of liver infection. Ad5-RGD failed to complete replicative cycle in hepatocytes, and transduced cells remained impermeable for propidium iodide, with a small fraction undergoing spontaneous apoptosis. In Ad5-RGD-immune mice, the virus neither killed KCs nor transduced hepatocytes, while in the setting of hepatic regeneration, Ad5-RGD enhanced liver transduction. The clinical and biochemical signs of hepatotoxicity correlated well with KC death, but not hepatocyte transduction. Real-time in vivo tracking for dynamic interactions between virus and host cells provides a better understanding of mechanisms underlying Ad-related hepatotoxicity.

## 1. Introduction

Adenoviruses (Ads) are among the most studied viral vectors for gene therapy and oncolytic virotherapy (OVT). However, hepatotoxicity is one of the principal obstacles to their widespread use. Tremendous efforts were applied to identify the factors that determine the Ad interactions with hepatocytes and immune cells in the liver. While these studies significantly improved our understanding of Ad-mediated liver toxicity, there are still blank spots in the area [1].

The Ad-related hepatotoxicity can be classified into early (capsid-dependent) and late (transcript-dependent) phases. The early toxicity develops within minutes–hours after systemic injection and is associated with a virus uptake by the liver macrophages (Kupffer cells, KCs), followed by KC death. The damaged KCs massively release proinflammatory factors into the blood, leading to the necrotic changes in the liver and a life-threatening systemic hemodynamic response [2,3,4,5,6], with the platelet-activating factor (PAF) being a key mediator of this acute toxicity [7]. As the KC depletion by Ads depends on the ability of the vector to escape endosomes, it was initially suggested that group B Ads would have less toxicity than group C, due to the delayed release from the endosomes [8]. However, later studies provided evidence for a differential early toxicity of group B Ads [9,10,11,12], highlighting the role of fiber shaft length in virus triggered KC death [10]. It still remains unclear whether KC death represents a host-driven mechanism of infection control or the virus-driven mechanism of escaping phagocytosis.

The late phase of toxicity is observed at the infection peak (48–72 h) and is thought to be triggered by hepatocyte transduction. The factors responsible for Ad tropism to hepatocytes are still debatable. While the roles of coagulation factor X (FX) and viral hexon in liver targeting are well established [13], there is no consensus regarding the importance of other serum factors, cellular receptors, as well as the involvement of leukocytes and platelets in the virus’ delivery to the hepatocytes [14,15,16]. Although several strategies have been implicated in preventing oncolytic Ads replication in normal tissues, the problem of transient liver transduction is still not completely resolved [17]. Only a few studies address the role of the immune cells in the clearance of infection and triggering of toxic side effects [18,19], and the fate of the virus-transduced hepatocytes is largely unknown.

Multiple Ad serotypes are currently under investigation in clinical trials, with Ad5RGD, Ad5/3, and Ad5/35 being among the most extensively studied vectors [20,21]. These viruses utilize different mechanisms to infect the cell. Ad5RGD, with the RGD-4C peptide inserted into the HI loop of the fiber knob, enters the cell via binding to the coxsackievirus and Ad receptor (CAR) as a primary high-affinity receptor, or via integrin receptors, mainly αVβ3 and αVβ5, in a CAR-independent manner [22]. The fiber chimeric Ad5/3, with the fiber knob domain derived from Ad3 [23], binds to the human DSG2 receptor as a primary high-affinity receptor [24], and to human CD46 as a low-affinity receptor [25], while the fiber chimeric Ad5/35, with the fiber shaft and knob domains derived from Ad35 [26], binds to the human CD46 [27]. To restrict viral replication to the cancer cells, oncolytic Ads are often designed with an E1AΔ24 modification, a 24-base pair deletion in a sequence encoding the conservative region 2 domain of the E1A protein, which binds SUMO-conjugase UBC9 [28], the stimulator of interferon genes [29], and the tumor-suppressor retinoblastoma protein (pRb) [30,31]. The differential cellular tropism of the Ad vectors not only affects the ability of a virus to infect tumor cells, but also influences virus–host interactions in the liver.

Deciphering the mechanisms responsible for Ad-related toxicity is crucial for the future of the virotherapy. The detailed characterization of the dynamic processes, such as the interactions of Ads with KCs, or the immune cell recruitment to sites of infection, requires special in vivo detection techniques. During the past decade, the rapid development of intravital microscopy (IVM) has provided a powerful tool for studying virus interactions with, and infection of, host cells in vivo. Here, using this method, we track the early and late stages of liver infection in mice upon administration of three oncolytic Ads: Ad5-Δ24-RGD, Ad5/3-Δ24, and Ad5/35-Δ24 (further referenced as Ad5-RGD, Ad5/3, and Ad5/35). Our intravital findings broaden the understanding of the virus–host interactions responsible for liver toxicity.

## 2. Materials and Methods

### 2.1. Recombinant Oncolytic Adenoviruses

Ad5-RGD-Δ24-E1B-p2A-Fluc, Ad5/3-Δ24-E1B-p2A-Fluc, Ad5/35-Δ24-E1B-p2A-Fluc, Ad5-RGD-Δ24-E1B-p2A-EGFP, Ad5/3-Δ24-DBP-p2A-EGFP, Ad5/35-Δ24-DBP-p2A-EGFP, and Ad-5RGD-Δ24-pIIIA-EGFP (Fib-L5) were constructed, rescued, amplified, purified, stored, and titered, as detailed in [32,33]. The labeling of the adenoviral constructs was achieved by adding 10 μg Alexa Fluo 647 NHS Ester (Thermo Fisher Scientific, Waltham, MA, USA) to 8 × 10^11^ viral particles (VP). After 20 min incubation at room temperature, the excess dye was removed by dialysis, using Slide-A-Lyzer 10K dialysis cassettes (Thermo Scientific) against 1L of storage buffer (5 mM Tris, 75 mM NaCl, 1 mM MgCl_2_, 5% sucrose (*w*/*v*), 0.005% Polysorbate 80, pH 8.0) at 4 °C overnight.

### 2.2. In Vitro Bioluminescent Assay

The AML12 murine hepatocytes (ATCC CRL-2254) were seeded onto 96-well white plates (2.5 × 10^4^ cells/well) and infected in suspension with serial dilutions of Ads. At 24 h post-infection (hpi) the cells were analyzed, as previously described [33].

### 2.3. Resazurin/AlamarBlue™ Cell Viability Assay

The AML12 cells (2500 cells/well) were seeded onto 96-well plates and infected in suspension with serial dilutions of Ads. Five days post-infection, the cells were analyzed as previously described [33].

### 2.4. Animals and Treatments

All of the animal studies were approved by the Animal Care Committee of N. I. Pirogov Russian National Research Medical University. The six to eight-week-old female BALB/c mice were obtained from Andreevka Animal Center.

For the bioluminescence, biochemical studies, flow cytometry (FC) and cytokine analysis, the mice received intravenously (i.v.) 7 × 10^8^ infectious units (IFU) of Fluc-expressing vectors (unless otherwise noted). To analyze the second dose transduction efficiency and toxicity, 7 × 10^8^ IFU of EGFP-expressing Ad5-RGD was followed by the same dose of Fluc-expressing Ad5-RGD at 72 h or 14 days after initial treatment. The animal body weights were measured on a daily basis. For studying virus capturing, 10^10^ of fluorescently labeled VPs were i.v. injected. The intravital imaging of hepatocyte infection and immune response was performed in mice injected i.v. with 1.6 × 10^9^ IFU of EGFP-encoding vectors. Neutrophil depletion and CD8+ T cell depletion were achieved by intraperitoneal (i.p.) injection of 250 μg anti-Ly6G (clone 1A8; BioXCell, West Lebanon, NH, USA) or 250 μg anti-CD8 (clone 53.6.7; BioXCell) antibodies, respectively; for each depleting antibody, two treatment schedules were tested: 24 h before or 1 h after Ad5-RGD administration. As a proper control, mice receiving 250 μg of rat anti-IgG2α (clone 2A3, BioXCell) were used. For the macrophage/monocyte depletion, animals were treated i.p. with 1 mg clodronate liposomes (Encapsula NanoSciences, Brentwood, TN, USA) 24 h before the virus challenge. PAF receptor antagonist ABT-491 (25 μg in 100 μL of PBS; Sigma-Aldrich, St. Louis, MO, USA) was i.v. injected 20–30 min prior to Ad5-RGD administration. To study liver infection and toxicity in the setting of regeneration, mice received 2 × 10^8^ IFU of *Fluc*-encoding Ad5-RGD at 72 h following hepatectomy or sham surgery.

Partial hepatectomies were performed, as previously described [34] with modifications. Briefly, the mice were anesthetized (Zoletil 50 mg/kg, Xylazine 5 mg/kg), shaved, and sanitized with 70% ethanol. An incision was made along the midline of the abdomen, the median lobe of liver was exposed, ligated with 4-0 silk suture and removed with scissors. As a proper control, sham-operated mice were used, which underwent the same procedures, except for the removal of a liver lobe, including sanitation, abdomen cut, liver lobes’ rotation, saline instillation, and wound closure by sutures.

### 2.5. Intravital Microscopy

The liver for the IVM was prepared as described elsewhere [35]. The host cells were stained by i.v. injection of fluorescently labeled antibodies: Ly6G BV421 (clone 1A8, 0.6 μg); CD31 BV421 (clone 390, 1 μg); CD45 BV421 (clone 30-F11, 1.4 μg); F4/80 AF488 (clone BM8, 2.5 μg); CD11b PE (clone M1/70, 0.6 μg) from Biolegend (San Diego, CA, USA); CD49b PE (clone DX5, 1.4 μg); CD8a eFluor660 (clone 53.67, 2.5 μg) from ThermoFisher Scientific. The adenoviral vector uptake by the liver cells was studied at the moment of, and within 30–90 min after, injection of the labeled virions (3–4 mice in a group; 5–11 fields of view for each animal) with the acquisition rate of 1.5 frame/min, using inverted confocal microscope Nikon A1R (Tokyo, Japan). Hepatocyte transduction and immune cell trafficking were assessed 24–72 h after i.v. injection of the EGFP-expressing vectors. The plasma membrane integrity of the liver cells was tested by exclusion of propidium iodide (PI; 50 μg in 100 μL PBS) injected i.v. at early (40 min) and late (24–72 h) stages of infection. After the liver IVM, the animals were sacrificed and the cut spleens were scanned ex vivo.

The post-processing analysis of the movies was performed in NIS Elements AR software (Nikon). To quantify the uptake of the adenoviral vectors by KCs, the binary layers were generated in F4/80 channel and converted into regions of interest (ROIs) for each individual frame. Then, mean fluorescence intensity (MFI) was measured in the obtained ROIs as a function of time. The percentage of PI-positive KCs in each individual frame was calculated, using a standard equation with the following parameters: total number of KCs—a number of objects under F4/80 binary layer; PI-positive KCs—number of objects under intersection of F4/80 and PI binary layers.

### 2.6. Bioluminescence Imaging

At the indicated time points, the mice treated with the Fluc-expressing adenoviral vectors were injected i.p. with 150 mg/kg of firefly D-luciferin (Perkin Elmer, Waltham, MA, USA) in PBS and allowed to rest for 10 min. The imaging was conducted using IVIS Spectrum CT imaging system (Perkin Elmer) and the photon emission values were calculated with Living Image 4.3 software.

### 2.7. Flow Cytometry

The leukocyte single cell suspensions from the livers were obtained, as described elsewhere [36]. After treatment with anti-CD16/CD32 antibodies (1/100, clone 93, Biolegend), the cells were stained for 30 min at 4 °C with combinations of the following antibodies (1/100): eFluor 660 anti-mouse CD8a (clone 53-6.7); eFluor 660 anti-mouse F4/80 (clone BM8) from ThermoFisher Scientific; Brilliant Violet 421 anti-mouse NK-1.1 (clone PK136); Alexa Fluor 488 anti-mouse Ly6C (clone HK1.4); PE anti-mouse F4/80 (clone BM8); PerCP anti-mouse Ly6G (clone 1A8); PE/Cy7 anti-mouse/human CD11b (clone M1/70); APC/Cy7 anti-mouse CD45 (clone 30-F11); and isotype controls from Biolegend. The data analysis was performed in Summit 5.2.0 (Beckman-Coulter, Fullerton, CA, USA). The distinct cells’ populations were gated, as shown in Appendix A.

### 2.8. Analysis of Alanine Aminotransferase (ALT), Aspartate Aminotransferase (AST), and Lactate Dehydrogenase (LDH) Levels in Plasma

Blood was collected with 2% EDTA solution through the cardiac puncture. The ALT and AST plasma levels were measured using the automated biochemical analyzer (BioChem FC-120, High Technology, North Attleboro, MA, USA) and commercial kits (HT-A306-120, HT-A309-120, High Technology), according to the manufacturer’s protocol. The LDH was measured with a LDH detection kit (HT-L336-600-120, High Technology), following the supplier’s protocol modified for a plate-reader (EnSpire 2300 Multilabel Reader, PerkinElmer).

### 2.9. Detection of Anti-Ad IgM/IgG

Total IgM and IgG ELISA assays (Thermo Fisher Scientific) were used to measure Anti-Ad IgM/IgG, as described by Xu et al. [37]. Briefly, anti-IgM (or anti-IgG) capturing antibodies from commercial kit were replaced with Ad-5RGD (3.0 × 10^10^ VP/mL). All of the subsequent procedures of ELISA assay were performed according to the manufacturer’s protocol. The plasma samples were 400-fold and 20-fold diluted in assay buffer for the detection of anti-Ad IgM and IgG, respectively.

### 2.10. Cytokine Analysis

The plasma levels of IFN-α, IFN-γ, TNF-α, IL-6, IL-4, IL-10, CCL2, CCL3, CCL4, CXCL9, CXCL10, VEGF, and GM-CSF were measured with 13-plex Mouse Cytokine Release Syndrome Panel (Biolegend), using MoFlo Astrios EQ Sorter (Beckman-Coulter). The data analysis was performed in the LEGENDplex Data Analysis Software Suite (Biolegend).

### 2.11. Statistics

The data analysis was performed in GraphPad Prism 8.0.1 (GraphPad Software, San Diego, CA, USA) using an unpaired *t*-test, the Mann–Whitney test, and one- or two-way ANOVA, followed by Tukey’s or Dunnett’s multiple comparisons tests.

## 3. Results

### 3.1. In Vivo Transduction of Hepatocytes Does Not Correlate with Toxicity

First, we compared the infection level and toxicity of the oncolytic vectors Ad5-RGD, Ad5/3, and Ad5/35 in vitro and in vivo. In the murine hepatocytes AML12, the transduction efficiency was much higher for Ad5-RGD than for the other two viruses, as measured by luciferase activity 24 hpi with *Fluc*-encoding constructs (Figure 1A). These results correlated well with the cytotoxicity at 120 h after infection (Figure 1B).

For studying the in vivo transduction, we gave mice i.v. an equal dose of Fluc-expressing vectors (7 × 10^8^ IFU/mouse) and the bioluminescence was measured. With this dose, the transduction rate in the liver reached a maximum at 48 hpi and, in agreement with the in vitro results, Ad5-RGD demonstrated two to five-fold higher Fluc expression levels than Ad5/3 and Ad5/35 (Figure 1C; Appendix A). As expected, all three of the viruses selectively infected hepatocytes, as shown by the liver IVM with EGFP-expressing viral constructs at 24 hpi (Appendix A).

Several approaches were used to assess the in vivo toxicity of the studied vectors. Consistent with the previous reports [4,5,7], we noted the transient signs of hyperacute distress (prostration, lethargy, acrocyanosis) within 10–30 min after the Ad5-RGD administration. The symptoms were never observed after the Ad5/35 injection, but they were even more severe following the Ad5/3 treatment. The plasma levels of ALT and AST were measured as an indicator of hepatocyte damage; additionally, LDH was analyzed as a marker of KC death [2,6,38]. Of note, the increase in the plasma enzyme levels was detected early after infection (12 h), but not at the infection peak (Appendix A). Based on these results, we selected the 12 h time point to compare hepatotoxicity of the studied vectors. The levels of the enzymes between the groups treated with different viral constructs were not proportional to the liver transduction rates. While all three of the viruses led to a moderate increase in ALT levels (Figure 1D), an elevation of AST was detected only in the Ad5-RGD- and Ad5/3-treated mice (Figure 1E). Additionally, the Ad5-RGD administration was associated with increased LDH plasma levels (Figure 1F). None of the viruses caused an elevation of the TNF-α or IL-6, the markers of a cytokine storm (Appendix A). Finally, we observed a trend (although non-significant) towards weight loss in the mice that received Ad5/3, in comparison to the other vectors (Appendix A). Collectively, the results suggest that the in vivo hepatotoxicity of oncolytic Ads is not related to the viral gene expression in hepatocytes, but is most likely associated with early virus interactions with liver cells.

### 3.2. Ad5-RGD and Ad5/3 Vectors Lead to Rapid Zeiosis of Kupffer Cells

To study the early events of infection, the liver was imaged intravitally while injecting fluorescently labeled virions. For this purpose, equal numbers of VP (10^10^ VP/mouse) of the various vectors were administered. As expected, for all of the viral constructs, the major part of the virions was rapidly taken up by KCs, with only minimal binding to endothelial cells (Appendix A). However, the accumulation level of the viral particles in KCs was lower for Ad5/35, than for Ad5-RGD and Ad5/3 (Figure 2A,B; Appendix A). Unfortunately, we were not able to capture interactions of the single virions with the hepatocytes, due to limitations in resolution of IVM technique.

Looking closely at the behavior of the Ad5-RGD- and Ad5/3-containing macrophages, we found signs of zeiosis, a type of membrane blebbing associated with apoptosis [39]. As early as 15–30 min after infusion, small surface membrane blebs and large dynamic membrane blebs were a common finding in KCs (Figure 2C; Appendix A). We also observed the formation of apoptotic membrane protrusions, which sometimes extended for more than 100 µm (Appendix A); in certain cases, the virions were trafficking inside the protrusions (Figure 2D; Appendix A). Moreover, the viral particles were released from the KCs into blood inside the membrane-associated vesicles (Figure 2E; Appendix A). Although we did not observe neutrophils capturing the floating virus, these cells were able to sweep out the Ad-containing macrophages (Figure 2F), and to uptake the virions associated with KC membranes (Appendix A).

Notably, the described changes in the macrophages behavior were never spotted after the Ad5/35 infusion, even for the cells that accumulated a number of virions that was comparable with the two other vectors. Consistent with these findings and the clinical signs of toxicity, 30–40% of KCs lost their plasma membrane integrity, as evidenced by the uptake of PI, cell-impermeant dye, administered at 40 min after the injection of Ad5-RGD and Ad5/3, while the Ad5/35 impact on cell viability was negligible (Figure 2G; Appendix A). These intravital observations confirm the established role of Ad-associated KC death in acute viral toxicity and provide evidence for the virus escape from the dying macrophages in membrane-bound vesicles and on migrating neutrophils.

### 3.3. Virus-Transduced Hepatocytes Are Cleared Non-Cytolytically

Next, we examined the fate of the virus-transduced hepatocytes and the immune response to the infection in the liver. These experiments were performed with Ad5-RGD, the vector that demonstrated the most profound liver transduction. First, we compared the number of transduced hepatocytes at 48–72 hpi of the two Ad5-RGD vectors, with EGFP expression driven by either early (*E1B*) or late (major late promoter, *MLP*) viral promoters. While the *E1B* promoter-controlled transgene expression resulted in diffuse liver staining, the EGFP-expression linked to the MLP promoter was an extremely rare finding (Figure 3A,B), pointing towards the abortive infection.

To estimate the role of immune cells in the infection control, leukocytes’ trafficking in the liver was studied by IVM at 24–72 hpi. The innate immune effectors (neutrophils, monocytes, NK-cells, platelets) did not interact with the EGFP-positive cells, although long-term contacts between the CD8+ T cells and the infected hepatocytes were found on a regular basis (Figure 3C; Appendix A). These interactions resembled an immunological synapse; however, we never observed any morphological change of the infected cells, despite the contacts often lasted for more than 30 min. The cellular membrane integrity of the viral-transduced cells was not compromised, as evidenced by the exclusion of i.v. injected PI at the peak of infection (48 h), or when the infection started to wane (72 h; Figure 3D). However, in less than 1% of the EGFP-expressing hepatocytes we were able to capture spontaneous apoptotic changes (Figure 3E; Appendix A). The obtained results suggest that hepatocyte infection by oncolytic Ad5-RGD is abortive and cleared non-cytolytically.

### 3.4. Immune Response to Adenovirus Infection in the Liver

To further investigate the potential role of the immune cells in the control of virus spread, we analyzed the liver leukocyte subpopulations by FC during early (12 h) and late (72 h) phase of Ad5-RGD infection (Figure 3F; Appendix A). In agreement with the previous reports [2,4,6] and our IVM data, the significant (>50%) loss of KCs was detected at 12 hpi. Notably, by 72 hpi, the number of the liver macrophages returned back to normal. The rapid recovery of the KC population is likely achieved by recruitment of monocytes, whose number is sharply increased early after infection in response to the “open status” of the KC niche [40].

The neutrophils and NK cells were also enrolled in the acute immune response to the virus, and the frequency of the liver NK cells was still elevated at 72 hpi. The CD8+ T cell numbers were unchanged at 12 hpi with a trend (*p* = 0.07) to an increase at 72 hpi (Figure 3F). Similar changes in the leukocyte counts were found 12 h after administration of Ad5/3, while Ad5/35 did not elicit a cellular response in the liver (Appendix A). The recruitment of the leukocytes to the liver early after infection with Ad5-RGD and Ad5/3 coincided with the increase in the plasma level of CXCL9, a chemokine primarily involved in the CD8+ T cell response (Appendix A). Of note, CXCL10, another important factor of lymphocyte recruitment, was elevated equally after injecting all three of the adenoviral vectors. The levels of other tested cytokines (IFN-α, IFN-γ, TNF-α, IL-6, IL-4, IL-10, CCL2, CCL3, CCL4, VEGF, GM-CSF) were unchanged at 12 hpi in all of the animal groups (data not shown).

The IVM observations, dynamics of liver leukocytes, and the systemic chemokine response suggested that the macrophages/monocytes (Figure 2A–E), neutrophils (Figure 2F and Figure 3F) and CD8+ T cells (Figure 3C; Appendix A) may influence the virus biodistribution and infection. To test these hypotheses, the liver infection was studied by bioluminescence after injection of *Fluc*-encoding Ad5-RGD into mice depleted of distinct leukocyte subsets. The elimination of the neutrophils or CD8+ T cells with anti-Ly6g or anti-CD8a antibodies (Appendix A) had no effect on the efficiency of the liver transduction by Ad5-RGD, although a trend (*p* = 0.1) towards decreased Fluc expression in the Ly6G-depleted mice was detected (Figure 3G; Appendix A).

In contrast, the macrophage/monocyte depletion by clodronate liposomes (Appendix A) resulted in a tremendous increase in the hepatocyte transduction (Appendix A), likely due to the virus redistribution from the phagocytes to other host cells. This result was not surprising, based on multiple previous reports [3,41,42,43], but we aimed to investigate another aspect of viral pathogenesis related to macrophages: whether virus-mediated KC death favors the infection of hepatocytes. In particular, Ad is known to trigger a profound release of PAF by KCs, which, in turn, leads to a sharp increase in vessel permeability [7]. We hypothesized that the PAF influx in the liver facilitated the virus trafficking to the hepatocytes through a sinusoid wall, contributing to more efficient liver transduction. The pretreatment with the PAF receptor antagonist ABT-491 completely abrogated clinical signs of acute toxicity after administration of both low (7 × 10^8^ IFU) and high (2 × 10^9^ IFU) doses of Ad5-RGD, however, it did not affect the level of liver infection (Figure 3H; Appendix A). In aggregate, these experiments show that the spread and clearance of the liver infection is not dependent on either neutrophils or CD8+ T cells, and that the macrophage-derived PAF does not facilitate the transduction of the hepatocytes.

### 3.5. Liver Infection and Toxicity in Preimmunized Hosts

In clinical trials, the systemic delivery of oncolytic Ads is typically scheduled as a series of intravascular injections, with an interval of 2 to 14 days between the doses [44,45,46]. In this regard, we sought to investigate the hepatic infection and toxicity following the repeated virus administration. The mice received 7 × 10^8^ IFU of *EGFP*-encoding Ad5-RGD, and then the same dose of *Fluc*-encoding Ad5-RGD was administered 72 h or 14 days later. These time points were selected to study the virus–host interactions at the early and late phases of adaptive immune response, characterized by different levels of anti-Ad specific IgM/IgG in serum (Appendix A). The bioluminescence imaging demonstrated that in both of the treatment schedules the virus failed to infect the liver (Figure 4A; Appendix A). The lack of liver transduction could be attributed to a more efficient antiviral response, in particular, a profound elevation of IFNs early after rechallenge (Figure 4B). A repeated virus injection after 14 days also stimulated a significant increase in the plasma levels of CXCL9/CXCL10 chemokines. Of note, animals of both of the groups receiving the second dose had neither clinical signs of acute toxicity nor elevation of transaminases at 12 h after the Ad5-RGD injection (Figure 4C).

The behavior of the fluorescently labeled virions administered as a second dose was also very different from that of the first dose. While the initial virus injection was accompanied by an even counterstaining of the vessels, that declined to undetectable levels within 5–10 min, the second dose of the labeled virions (administered either after 72 h or 14 days) appeared as aggregates circulating in the blood for more than 30 min. These viral aggregates (≈1–2 µm in diameter) often colocalized with the platelets, especially for the repeated dose injected after 14 days (Appendix A; Appendix A). Of note, the repeated dosing at 72 hpi resulted in a very limited virus uptake by the liver macrophages, but more profound Ad assimilation in the marginal zone of the spleen, as compared to the initial dosing or the second dose administered after 14 days (Figure 4D; Appendix A; Appendix A). More importantly, in both of the treatment schedules, the second dose of Ad5-RGD did not affect the KC viability, as opposed to the first dose effect (Figure 4E). These findings indicate that the biodistribution of Ads is dramatically affected by the preceding virus treatment and the interval between the doses.

### 3.6. Adenovirus Infection in Regenerating Liver

Oncolytic viruses can be used as an adjuvant therapy after surgical resection of the tumor, including hepatic cancer. Although it is known that the sites of tissue repair provide favorable conditions for the replication of oncolytic herpes simplex virus (HSV), vaccinia virus (VV), and vesicular stomatitis virus [17], the infection of Ad in the setting of liver regeneration has not yet been studied. At 72 h after 30% hepatectomy or sham surgery, the animals received 2 × 10^8^ IFU of Fluc-expressing Ad5-RGD. Despite significant variability in bioluminescence results, the hepatectomized mice demonstrated higher liver transduction rates as compared to the groups with no surgery or sham surgery (Figure 4F; Appendix A). With this dose, the increased infection of the regenerating liver was not associated with either clinical signs of toxicity or body weight loss (Appendix A).

## 4. Discussion

To evaluate the role of capsid proteins versus vector gene expression in mediating liver toxicity, we studied three oncolytic viruses displaying different patterns of biodistribution: Ad5-RGD, with a high uptake rate by KCs and a high level of hepatocyte transduction; Ad5/3, with a high uptake rate by KCs and a low level of hepatocyte transduction; and Ad5/35, with a low uptake rate by KCs and a low level of hepatocyte transduction. This work does not address the mechanisms responsible for the differential uptake of adenoviral constructs by KCs and hepatocytes. Based on the previous studies, it is likely that superior in vitro infectivity of Ad5-RGD over the two other vectors stems from the differences in expression levels of the viral entry receptors (CAR, DSG2, CD46) on murine hepatocytes [24,47]. In vivo, liver transduction is mainly governed by the ability of the virion to bind with FX [48,49,50], while the virus uptake by the macrophages is at least partially mediated by scavenger receptors [11,16] and the complement receptor Ig-superfamily (CRIg) [51].

As early as 15–30 min post Ad5-RGD and Ad5/3 infusion (but not Ad5/35) we observed dramatic changes in the behavior of the virus-bound KCs. Previously, similar dynamic morphological changes of the cellular membrane (referred to as zeiosis or cell boiling) were documented in vitro in the macrophages upon influenza virus infection [52], and in epithelial cells after adding VV [53]. To the best of our knowledge, this is the first report on virus-induced zeiosis in vivo. In agreement with the previous studies [2,10,11,38,51], we observed a gross change in the plasmalemmal permeability of KCs following the uptake of Ad5-RGD and Ad5/3. Of note, zeiosis is typically associated with apoptosis, while a drastic increase in membrane permeability is a hallmark of necrosis [39]. KC death requires the virus to exit to the cytosol [10,38,54]; it is triggered by interferon-regulatory factor 3 (IRF3), but is independent of the known principal mediators of both apoptotic and necrotic cell-death programs [38]. The molecular mechanisms producing this atypical form of cell death remain to be determined.

Our data on KC zeiosis shortly after Ad treatment sharply differ from another IVM study, where no large-scale alterations in the KC morphology or behavior were found during a 4 h observation period [51]. It is possible that the conflicting results are due to the difference in mouse strains, as KCs in BALB/c mice (used in the current study) are characterized by much more effective Ad uptake, than in C57bl/6 mice [43].

The reduced hepatotoxicity of Ad5/35, as compared to the other *E1A*Δ24-harboring viruses shown here, is consistent with the previous reports on Ad5, Ad3, Ad35, and Ad5/35 vectors with wild-type E1A, in which the Ad5/35 was less toxic than the other viruses [9,10,12,14,55,56,57,58]. These data, coupled with a rapid development of the toxic signs, indicate that the hepatotoxicity of the oncolytic viruses is capsid-dependent rather than transcript-dependent. The improved safety of Ad5/35 is most likely associated with the Ad35 fiber that is shorter than the Ad5 fiber [32,33]. Previously, the Ad5/35 vectors, with the whole fiber derived from Ad35, demonstrated less toxicity, while the chimeric vectors containing only the Ad35 fiber knob were equally effective with parental Ad5 in killing KCs [10,11,41]. Consistent with other reports [10,55,59,60], we found that, in general, the KCs captured less Ad5/35 than Ad5-RGD, however, even the KCs with high Ad5/35 uptake rates, sporadically detected throughout the liver, also had no signs of cell damage. These results suggest that a lack of KC depletion by Ad5/35 is not solely due to less efficient virus binding, but likely reflects the differences between Ad5/35 and the two other vectors in intracellular trafficking.

Di Paolo et al. suggested that KC death is a “defensive suicide”, that limits pathogen dissemination [38]. On the other hand, the described phenomenon could be a mechanism that the virus evolved to escape phagocytosis. The virus release in membrane-associated vesicles and the virus hand-off from the blebbing macrophages to the neutrophils support this possibility, although further studies are needed to test the hypothesis. We assumed that the previously described virus-triggered PAF release by the macrophages could help the virus to reach the hepatocytes by increasing the permeability of the sinusoid walls. However, pretreatment with a PAF receptor antagonist did not affect the transduction efficiency in the liver. A profound increase in the number of transduced hepatocytes in macrophage-depleted animals, shown in this and previous studies [3,41,42,43], further indicates that the macrophages prevent rather than facilitate hepatocyte targeting.

In agreement with earlier reports [19,61], we observed the increased frequencies of liver neutrophils in the early stages of Ad5-RGD and Ad5/3 infection, along with their long-lasting contacts with virus-containing KCs and the uptake of F4/80+ cellular fragments. These results point towards a neutrophil impact on the clearance of the dying KCs, similar to what has been shown for the Ad-containing marginal-zone splenic macrophages [62]. In contrast, at the infection peak, the neutrophil infiltration resolved and these cells did not interact with the virus-transduced hepatocytes. Furthermore, the anti-Ly6G antibodies neither enhanced nor prolonged the liver transduction, indicating that the neutrophils were not required for the clearance of infected hepatocytes.

Opposite to neutrophils, the T lymphocytes readily interacted with the transduced hepatocytes at 24–72 hpi. IVM revealed the CD8+ T cells flattened against the target cells, suggesting that a temporary synapse could function to mediate the cytotoxic interactions. Moreover, we observed elevated plasma levels of chemokines essential for lymphocyte recruitment [63,64] at 12 hpi, and a trend (*p* = 0.07) for increased CD8+ T cell frequencies in the liver at 72 hpi. However, the depletion of CD8+ T cells had no effect on the liver transduction rate, indicating that these cells do not play a major role in the elimination of infected hepatocytes. The observed long-lasting contacts could also reflect T cells being primed by the infected hepatocytes. The antigen presentation by the liver parenchymal cells leads to suboptimal T cell activation. This tolerogenic response is necessary to prevent hyperimmune reactions in the liver that is continuously exposed to multiple microbial constituents and nonpathogenic food antigens [65,66]. Further studies are needed to elucidate the role of the transduced hepatocytes in eliciting systemic antiviral and antitumor T cell responses during OVT. Despite the recruitment of the NK cells to the liver, we failed to reveal their direct contact with the transduced hepatocytes. Along with the cytolytic effect, the NK cell is one of the major producers of IFN-γ [67], and, therefore, the absence of NK-cell interactions with hepatocytes does not exclude their role in limiting the liver infection.

We also explored the possibility of hepatocyte clearance through the virus-mediated lysis. The comparison of transduction rates between the two viral constructs, with EGFP-expression under control of the early or late viral promoter clearly demonstrated that the virus failed to complete the life cycle in murine hepatocytes. Most of the transduced cells were morphologically unchanged at 24–72 hpi, with a small fraction demonstrating signs of apoptosis. Of note, their cellular membranes remained impermeable for PI, as opposed to virus-bound KCs. These data correlated well with the normal rates of ALT, AST, and LDH during the liver transduction peak.

It should be noted though, that the rodent models have certain limitations for studying Ads, and, therefore, the lack of productive liver infection cannot be directly extrapolated to humans. Furthermore, transgene-mediated toxicity may become a dose-limiting factor in regenerating liver. The hepatocytes are notorious for high proliferative potential [68], and increased levels of ribonucleotide reductase in dividing cells is one of the mechanisms responsible for an increased susceptibility of cells to oncolytic viruses [17]. Enhanced Ad binding to, and transduction of, mitotic cells were previously shown in vitro [69], and our results demonstrate an increased hepatic transduction in the setting of liver regeneration. Of note, oncolytic HSV leads to severe toxicity at early stages of hepatic regeneration [70]. Although with a low dose of Ad5-RGD (2 × 10^8^ IFU) we did not see any side effects in hepatectomized mice, the potential risk of the enhanced liver transduction should be taken into consideration when combining OVT with liver surgery.

Preexisting immunity is another game-changing factor for the efficiency and toxicity of Ad-based therapies. The anti-Ad neutralizing antibodies may arise from natural infections or previous cycles of virotherapy. While some reports show significantly diminished toxic effects following the second dose [5,71,72], other studies do not confirm these findings, or even describe the enhanced toxicity as compared to the initial treatment [73,74]. To test if the conflicting data could be explained by the timing of the Ad rechallenge, we delivered the second dose of Ad5-RGD at 72 h and 14 days after first dosing. Although liver transduction, KC damage, and transaminitis were absent in both of the experimental schedules, our data suggest that similar outcomes could be mediated by the different mechanisms. The KC uptake of the virus re-administered at 72 hpi was dramatically decreased as compared to the initial treatment. Such a remarkable change in virus biodistribution is probably the consequence of KC depletion after the first dose. Although FC demonstrates that the KC population is replenished by 72 hpi, bone marrow-derived macrophages have low expression levels of the scavenger receptor, a key mediator of virus uptake [75]. Conceivably, the inability of the newly formed KCs to efficiently clear Ad from the circulation leads to more prominent virus accumulation in the spleen. In contrast, re-dosing at 14 days resulted in only a marginal decline in the Ad5-RGD uptake by KCs, in comparison to the virus-naïve mice, however, the virus failed to kill the cells. Possibly, virus-specific IgG prevents KC depletion through an alteration of the Ad intracellular trafficking, in particular, by mediating TRIM21-dependent degradation of the capsid [76]. Of note, the TRIM21 interaction with the antibody-bound pathogen triggers a proinflammatory response [13], that could explain a marked increase in the plasma levels of IFN-α, IFN-γ, CXCL9, and CXCL10 following the second treatment in mice protected with anti-Ad IgG. The virion aggregation in the blood shown in the current study is also likely to be the consequence of Ad opsonization by the virus-specific antibodies. These findings, as well as the established roles of neutralizing antibodies in complement activation, inhibition of receptor binding, and blockage of Ad-FX complex [76,77], can explain the lack of transgene expression in the virus-pretreated animals.

Previous reports provided conflicting data on the ability of Ads to interact with platelets, and the role of intrahepatic clotting in virus delivery to KCs [14,15,16]. Our results suggest that, in naïve animals, platelets do not take up Ads, but these interactions do happen in preimmunized mice, presumably due to a redirection of the opsonized virions/aggregates to as yet unidentified receptors on the platelets. It should be noted though, that clot formation in liver sinusoids was never observed, even upon repeated dosing.

## 5. Conclusions

In summary, this work describes several aspects of Ad interaction with KCs and hepatocytes, as well as the impact of innate and adaptive immune responses on liver transduction and toxicity. Our results indicate that hepatocyte infection by oncolytic Ads is: (i) self-limiting and cleared non-cytolytically; (ii) enhanced in regenerating liver; (iii) not accompanied by inflammation and toxicity. In contrast, the capsid-mediated toxicity: (i) represents a potential issue for the safety of OVT and gene therapy; (ii) correlates with rapid KC zeiosis and innate immune response; (iii) is not observed in preimmunized hosts. Real-time tracking for dynamic interactions between Ads and the host cells can help improve the efficiency and safety of OVT.

## Figures and Tables

**Figure 1 biomedicines-10-01697-f001:**
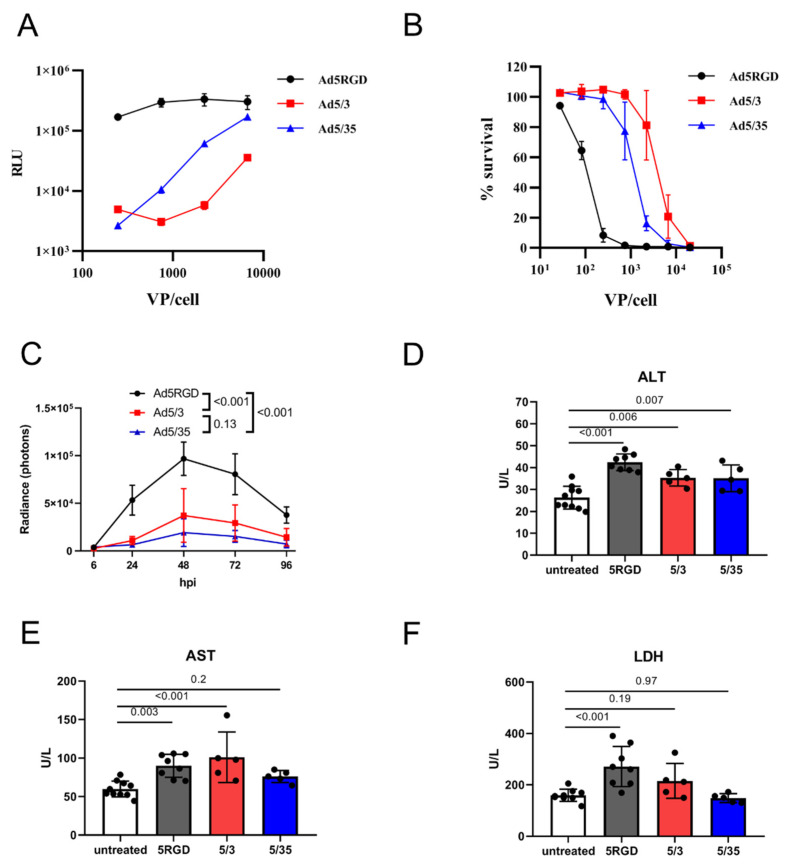
**In vivo transduction of hepatocytes does not correlate with toxicity.** (**A**). Transduction rates of Fluc-expressing adenoviral vectors in murine hepatocytes (AML12) measured by bioluminescent assay at 24 hpi; (**B**). Cytotoxicity of adenoviral vectors in murine hepatocytes (AML12) measured by AlamarBlue™ cell viability assay at 120 h after infection. For (**A**,**B**), the results of two independent experiments are plotted (mean ± SD); (**C**) In vivo liver transduction rates of i.v. injected Fluc-expressing adenoviral vectors (7 × 10^8^ IFU) measured by bioluminescence at indicated time points (*n* = 5; mean ± SD; *p*-values are shown on graph; two-way ANOVA followed by Tukey’s multiple comparisons test); (**D**–**F**). Plasma levels of ALT (**D**); AST (**E**); and LDH (**F**) measured at 12h after i.v. administration of adenoviral vectors (7 × 10^8^ IFU; mean ± SD; *p*-values are shown on graph; one-way ANOVA followed by Dunnett’s multiple comparisons test).

**Figure 2 biomedicines-10-01697-f002:**
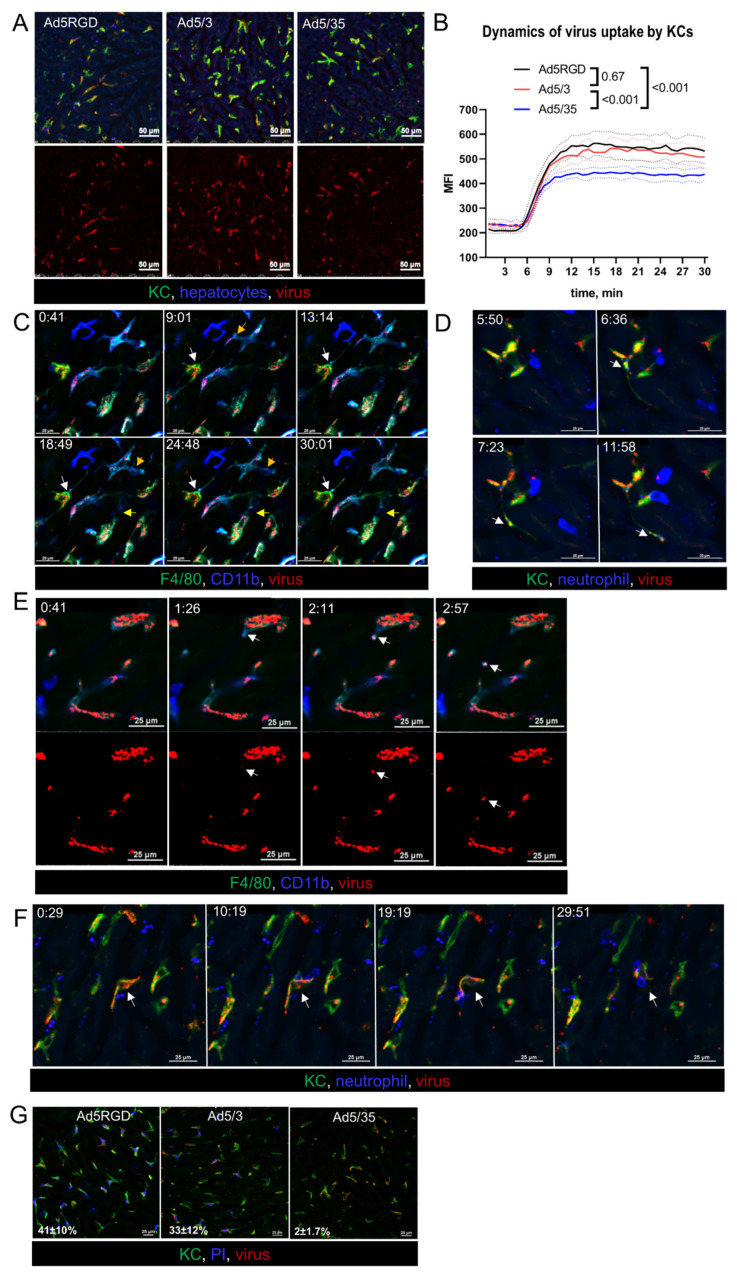
**Ad5-RGD and Ad5/3 vectors lead to rapid zeiosis of Kupffer cells.** (**A**,**B**). Representative images (**A**) and quantification (**B**) of adenoviral vectors’ uptake by KCs during 30 min after i.v. injection of 10^10^ VP labeled with AF647 (*n =* 4; mean ± SEM; *p*-values are shown on graph; two-way ANOVA followed by Tukey’s multiple comparisons test); (**C**)**.** Dynamic membrane bleb formation (yellow arrows) and apoptotic protrusion (white arrows) in virus-bound KCs; (**D**)**.** Virus trafficking (arrows) in KC apoptotic protrusion; (**E**)**.** Budding of virus-containing membrane-associated vesicle (arrows) from the surface of KC; (**F**). Interaction of neutrophils with virus-bound KC (arrows). For (**C**–**F**) acquisition time is shown as min:sec.; (**G**). Permeability of KCs for i.v. injected propidium iodide 40 min after i.v. injection of 10^10^ VP labeled with AF647. Percentage of PI-positive cells is shown as mean ± SD (*n =* 3).

**Figure 3 biomedicines-10-01697-f003:**
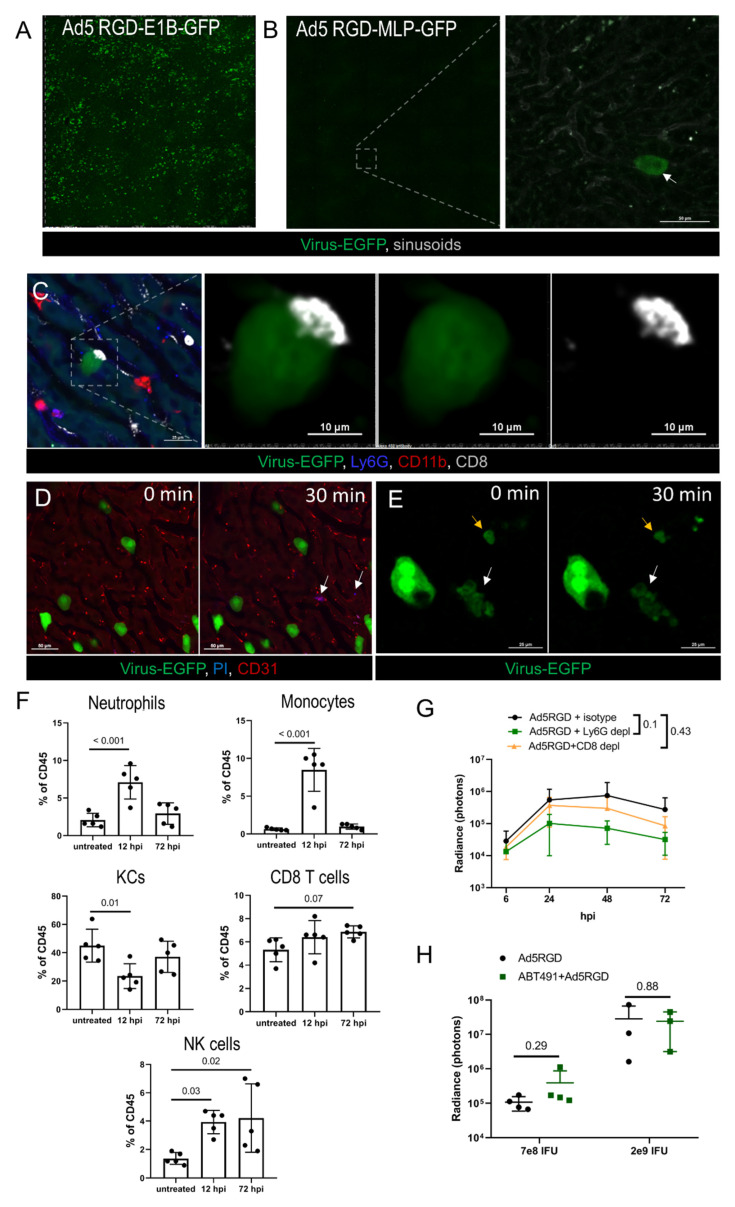
**Immune response and clearance of liver infection.** (**A**,**B**). IVM-images of the liver at 72 hpi of Ad5-RGD vectors (1.6 × 10^9^ IFU) with EGFP expression under control of early (**A**) or late (**B**) viral promoter. Arrow indicates an extremely rare finding—EGFP expression driven by late promoter; (**C**). Long-term interaction of CD8+ T cell with virus-transduced hepatocyte at 24 hpi; (**D**). Infected hepatocytes (72 hpi) before and 30 min after i.v. injection of propidium iodide (PI). Arrows show PI signaling outside virus-transduced hepatocytes; (**E**). Apoptotic changes of virus-transduced hepatocyte (white arrow) recorded during 30 min; yellow arrow depicts apoptotic bodies; (**F**). Liver leukocyte subsets 12 and 72 hpi of 7 × 10^8^ IFU Ad5-RGD (flow cytometry; mean ± SD; *p*-values are shown on graph; one-way ANOVA followed by Dunnett’s multiple comparisons test); (**G**). Ad5-RGD (7 × 10^8^ IFU) liver transduction rates measured by bioluminescence in animals treated with anti-Ly6G (*n =* 5), anti-CD8 (*n =* 5) antibodies or isotype control (*n =* 4; mean ± SD; *p*-values are shown on graph; two-way ANOVA followed by Dunnett’s multiple comparisons test); (**H**). Liver transduction rates measured by bioluminescence at 24 h after injection of Ad5-RGD (7 × 10^8^ IFU or 2 × 10^9^ IFU) into mice with or without ABT-491 pretreatment (mean ± SD; *p*-values are shown on graph; unpaired *t*-test).

**Figure 4 biomedicines-10-01697-f004:**
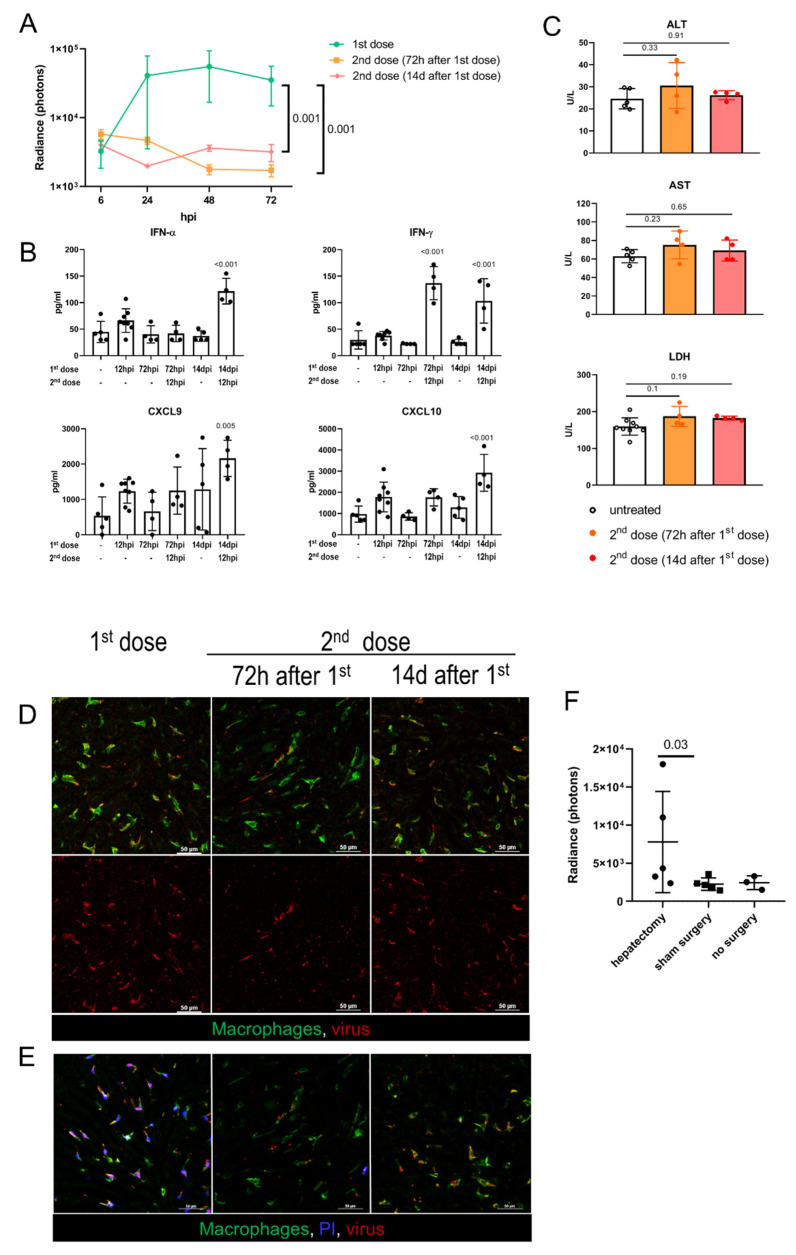
**Adenovirus infection and toxicity in preimmunized hosts and in regenerating liver.** (**A**). Ad5-RGD-Fluc (7 × 10^8^ IFU) liver transduction rates measured by bioluminescence in naïve mice (first dose, *n =* 5) and mice pretreated with 7 × 10^8^ IFU Ad5-RGD-EGFP 72 h (second dose (72 h after 1st dose); *n =* 5) or 14 days (second dose (14d after first dose); *n =* 3) before rechallenge (mean ± SD; *p*-values are shown on graph; two-way ANOVA followed by Dunnett’s multiple comparisons test); (**B**)**.** Plasma levels of selected cytokines in different treatment schedules (mean ± SD; *p*-values are shown on graph; one-way ANOVA followed by Dunnett’s multiple comparisons test); (**C**). Plasma levels of ALT, AST, and LDH measured at 12 hpi of the repeated Ad5-RGD dose (7 × 10^8^ IFU) re-administered 72 h or 14 days following the initial dose (mean ± SD; *p*-values are shown on graph; one-way ANOVA followed by Dunnett’s multiple comparisons test); (**D**). Representative images of the second dose uptake by KCs 30 min after injection of AF647-labeled Ad5-RGD (10^10^ VP); (**E**)**.** KC viability 40 min after injection of initial or repeated dose of AF647-labeled Ad5-RGD assessed by exclusion of i.v. administered propidium iodide (PI). (**F**). Liver transduction rates measured by bioluminescence at 24 h after injection of Ad5-RGD (2 × 10^8^ IFU) into animals underwent hepatectomy or sham surgery 72 h before virus challenge (mean ± SD; *p*-values are shown on graph; Mann–Whitney test).

## Data Availability

All original data of the study are presented in the article and queries can be directed to the corresponding author.

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
