# Peer review of "In Vivo Tracking for Oncolytic Adenovirus Interactions with Liver Cells"

_biomedicines, 2022, doi:10.3390/biomedicines10071697_

Round 1
Reviewer 1 Report
In this report the authors proposed an in vivo tracking system for oncolytic adenovirus (based on Ad5, Ad5/3, and Ad5/35, previously demonstrated to sequestrate in the liver and interact with hepatic cells) with hepatocytes of mice livers and in vitro in with AML12 cells in cell culture. The adenoviral constructs were labelled directly with Alexa Fluo 647 NHS Ester prior to use. The authors checked the adenoviral toxicity
and performed intravital microscopy to track adenovirus capsids. The report is accompanied with supplemental videos of the microscopy observations of the study. An interesting finding of this study is
the zeiosis, the first phase of cell apoptosis, of the adenovirus infected cells.
Author Response
We thank the Reviewer for the positive response.
Reviewer 2 Report
The article by Victor A. Naumenko et. al. entitled “In vivo tracking for oncolytic adenovirus interactions with liver cells” describes the oncolytic adenovirus interactions with liver cells in vivo, however, there are the following concerns that the authors should address.
1. The Authors are required to write the aim and objectives of the study in the abstract
2. The authors should provide their justification and relevance of the study. This will help the readers to understand the importance of the paper.
Relevant articles in the field such as J Virol. 2004 May;78(10):5368-81. doi: 10.1128/jvi.78.10.5368-5381.2004; J Exp Clin Cancer Res 35, 74 (2016); J Immunother Cancer. 2020 Aug;8(2):e001046. doi: 10.1136/jitc-2020-001046, may be discussed to improve the paper.
Author Response
- We included the aim of the study into the abstract:
Here by using intravital microscopy, we aimed to follow the infection and immune response in mouse liver from first minutes up to 72 h post intravenous injection of three Ads carrying delta-24 modification (Ad5-RGD, Ad5/3, and Ad5/35)
- We highlighted the importance of studying the mechanisms of Ad-related liver toxicity in Introduction:
Deciphering the mechanisms responsible for Ad-related toxicity is crucial for the future of the virotherapy.
- In our manuscript, we discuss the results obtained by Shayakhmetov et al. (doi: 10.1128/jvi.78.10.5368-5381.2004). As for a recent report by Flickinger et al. (doi: 10.1136/jitc-2020-001046), we don’t think it is relevant for the current paper as it describes toxicity after local (intramuscular) Ad administration, while our work is focused on systemic delivery.